# Microfluidic-based mini-metagenomics enables discovery of novel microbial lineages from complex environmental samples

Feiqiao Brian Yu[1,2], Paul C Blainey[3], Frederik Schulz[4], Tanja Woyke[4], Mark A Horowitz[1], Stephen R Quake[2,5,6]*

[1]Department of Electrical Engineering, Stanford University, Stanford, United States; [2]Department of Bioengineering, Stanford University, Stanford, United States; [3]MIT Department of Biological Engineering and Broad Institute of Harvard and MIT, Cambridge, United States; [4]Department of Energy Joint Genome Institute, Walnut Creek, United States; [5]Chan Zuckerberg Biohub, San Francisco, United States; [6]Department of Applied Physics, Stanford University, Stanford, United States

**Abstract** Metagenomics and single-cell genomics have enabled genome discovery from unknown branches of life. However, extracting novel genomes from complex mixtures of metagenomic data can still be challenging and represents an ill-posed problem which is generally approached with ad hoc methods. Here we present a microfluidic-based mini-metagenomic method which offers a statistically rigorous approach to extract novel microbial genomes while preserving single-cell resolution. We used this approach to analyze two hot spring samples from Yellowstone National Park and extracted 29 new genomes, including three deeply branching lineages. The single-cell resolution enabled accurate quantification of genome function and abundance, down to 1% in relative abundance. Our analyses of genome level SNP distributions also revealed low to moderate environmental selection. The scale, resolution, and statistical power of microfluidic-based mini-metagenomics make it a powerful tool to dissect the genomic structure of microbial communities while effectively preserving the fundamental unit of biology, the single cell.

*For correspondence: quake@stanford.edu

## Introduction

Advances in sequencing technologies have enabled the development of shotgun metagenomics and single-cell approaches to investigate environmental microbial communities. These studies revealed many previously uncharacterized genomes (*Brown et al., 2015*; *Eloe-Fadrosh et al., 2016*; *Kashtan et al., 2014*; *Rinke et al., 2013*), increasing the total number of sequenced microbial genomes to more than 50,000 (Joint Genome Institute's Integrated Microbial Genomes database, accessed December 1, 2016). However, the majority of environmental microbial diversity remains uncharacterized due to limitations in current techniques. Conventional shotgun metagenomic sequencing offers the ability to assemble genomes from a single heterogeneous sample, but is effective only if the complexity of the sample is not too great (*Howe et al., 2014*). Furthermore, it is often difficult to separate contigs belonging to closely related organisms because techniques designed to resolve these differences, such as tetranucleotide analysis, depend on ad hoc assumptions about nucleotide usage (*Dick et al., 2009*). It is possible to perform rigorous assemblies from independent single-cell genome amplifications but at the expense of lowering throughput (*Rinke et al., 2014*; *Blainey, 2013*). In addition, when performed in plates, single-cell sequencing approaches are

typically expensive and laborious, although microfluidic approaches have helped to alleviate that limitation (*Blainey et al., 2011*).

We report here a new mini-metagenomics approach which combines the advantages of shotgun and single-cell metagenomic analyses. This approach uses microfluidic parallelization to separate an environmental sample into many small sub-samples containing 5–10 cells, significantly reducing complexity of each sub-sample and allowing high quality assembly while enabling higher throughput than typical single-cell methods. Although each sub-sample contains a limited mixture of several genomes, single-cell resolution is regained through correlations of genome co-occurrence across sub-samples, which in turn enables a rigorous statistical interpretation of confidence and genome association.

We validated this approach using a synthetic mixture of defined microbial species and then applied it to analyze two hot spring samples from Yellowstone National Park. Among 29 genomes larger than 0.5 Mbps, most belong to known bacterial and archaeal phyla but represent novel lineages at lower taxonomic levels; three genomes represent deeply branching novel phylogenies. Functional analysis revealed different metabolic pathways that cells may use to achieve the same biochemical process such as nitrogen and sulfur reduction. Using information associated with genome occurrence across sub-samples, we further assessed abundance and genome variations at the single-cell level. Our analyses demonstrate the power of the mini-metagenomic approach in deconvolving genomes from complex samples and assessing diversity in a mixed microbial population.

## Results

### Evaluating microfluidic-based mini-metagenomics using a mock community reveals improved whole genome amplification

Microfluidic-based mini-metagenomics begins with microfluidic partitioning of each environmental sample randomly into 96 sub-samples with 5–10 cells per sub-sample (*Figure 1A*). 96 lysis and MDA (Multiple Displacement Amplification) reactions are performed in independent chambers of an automated and commercially available Fluidigm C1 Integrated Fluidic Circuit (IFC) (*Figure 1B*, *Figure 1—figure supplement 1*), which significantly reduces the time and effort required to perform these reactions. The C1 IFC has been used previously for mammalian single-cell RNA-seq and genome analysis experiments (*Wu et al., 2014*; *Pollen et al., 2014*; *Treutlein et al., 2014*; *Gawad et al., 2014*). The hardware, including the microfluidic circuit, was not altered for the mini-metagenomic experiments, but we did adapt a new reagent kit and designed scripts and protocols for amplifying genomic DNA from microbial cells. After whole-genome amplification, DNA from each sub-sample is harvested into a 96 well plate, and all subsequent wet lab steps are performed on the bench top. Sequencing libraries are prepared with distinct barcodes labeling DNA derived from each sub-sample and sequenced on the Illumina Nextseq platform (*Figure 1C*). Because of the small volumes used for microfluidic reactions, precious microbial samples and/or low cell concentrations that may not yield enough DNA for shotgun metagenomics can still be analyzed using this process. Another advantage of using a microfluidic platform is the enclosed reaction environment that limits potential contamination often observed with low input MDA reactions in well plates, a problem that, with other approaches, requires sophisticated protocols for contamination prevention and removal (*Woyke et al., 2011*). In addition, smaller MDA reaction volumes (~300 nL) are associated with lower gain and less amplification bias, which effectively improves coverage uniformity of amplified genomes (*de Bourcy et al., 2014*) (Materials and methods).

Sequence data is processed through a custom bioinformatics pipeline that takes advantage of information encapsulated in distinct yet related sub-samples (*Yu, 2017*). Reads from each sub-sample are trimmed and assembled into sub-sample contigs (*Figure 1D*), creating genome subassemblies. Since cells representing the same genome may appear in multiple sub-samples, overlapping genome subassemblies can be identified. Therefore, combined assembly of sub-sample reads and contigs results in longer mini-metagenomic contigs from which more meaningful biological information can be inferred (*Figure 1E*). With this mini-metagenomic approach, cells from the same phylogenetic groups are randomly partitioned into different sub-samples, providing a physically defined approach to bin metagenomic contigs. Aligning reads from each sub-sample to mini-metagenomic contigs enables us to determine the co-occurrence pattern of each contig (*Figure 1F,G*). Based on

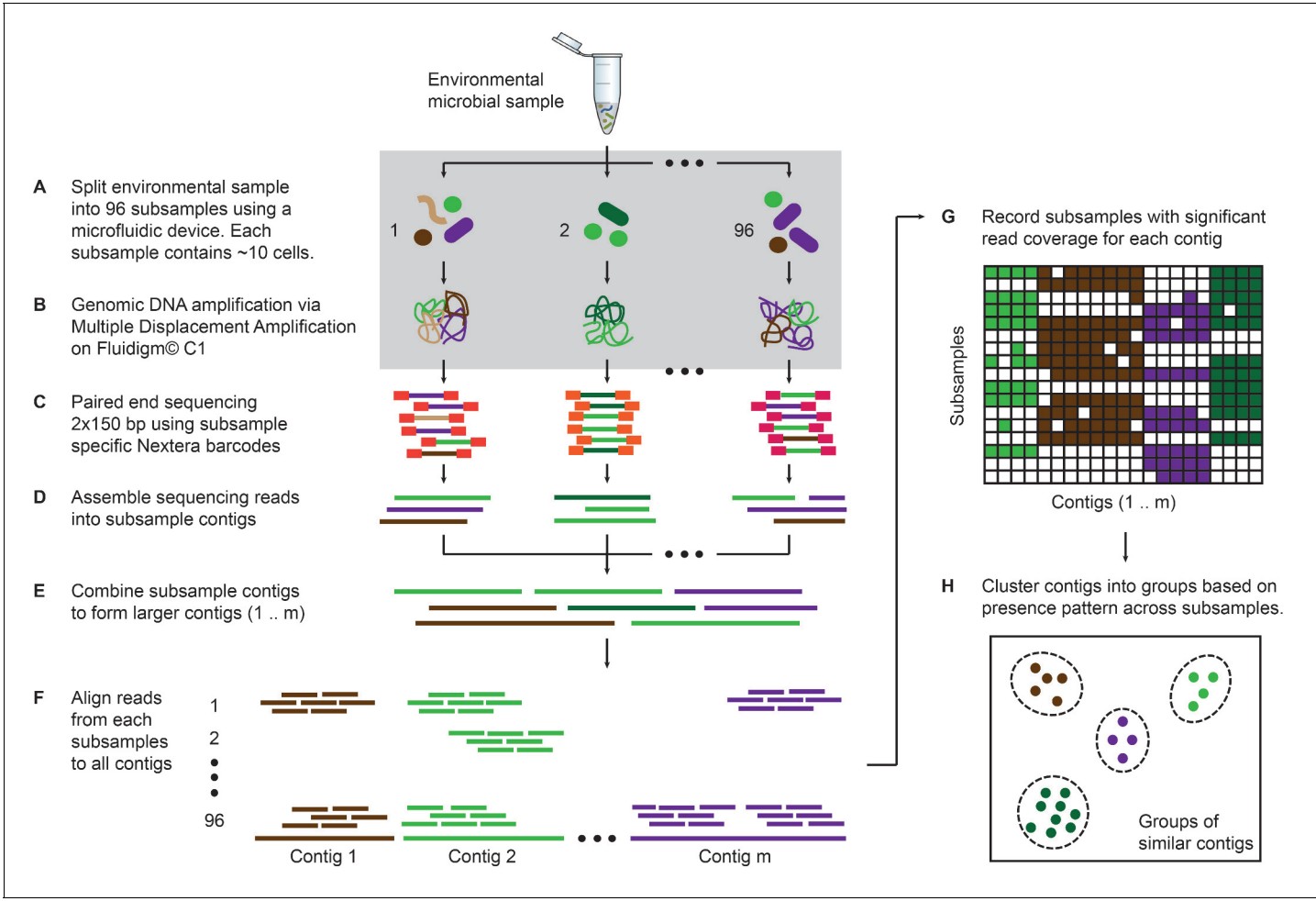

**Figure 1.** Microfluidic-based mini-metagenomics pipeline. (**A**) An environmental microbial sample is loaded onto a Fluidigm C1 IFC at the appropriate concentration so that cells are randomly dispersed into 96 microfluidic chambers at 5–10 cells per chamber. (**B**) Lysis and MDA are performed on the microfluidic device to generate 1–100 ng genomic DNA per sub-sample. (**C**) Nextera libraries are prepared from the amplified DNA off-chip and sequenced using 2 × 150 bp runs on the Illumina NextSeq platform. (**D**) Sequencing reads from each sub-sample are first assembled independently, then (**E**) sub-sample contigs are combined to form longer mini-metagenomic contigs. Contigs longer than 10 kbp are processed in the following steps. (**F**) Reads from each sub-sample are aligned to mini-metagenomic contigs > 10 kbp. (**G**) An occurrence map is generated, demonstrating the presence pattern of each contig in all sub-samples based on coverage. (**H**) Finally, contigs are binned into genome clusters based on a pairwise *p* value generated from co-occurrence information. Steps enclosed in the gray rectangle (**A, B**) are performed on the Fluidigm C1 IFC. Step C is carried out in 96 well plates. Steps D to H are performed in silico.

The following figure supplements are available for figure 1:

**Figure supplement 1.** Details of the mini-metagenomic experimental steps performed on the Fluidigm C1 microfluidic IFC.

**Figure supplement 2.** Performance of microfluidic-based mini-metagenomic amplification.

**Figure supplement 3.** Mini-metagenomics performance on mock communities.

presence patterns across sub-samples, a *p* value for each pair of contigs is computed based on Fisher's exact test (Materials and methods), where a small *p* is interpreted as an indicator that two contigs belong to cells of the same genome. Finally, co-occurrence based *p* values are used as a pairwise distance metric for sequence-independent contig clustering (*Figure 1H*; Materials and methods).

We tested performance of the mini-metagenomic amplification on the Fluidigm C1 IFC using a mixture of five bacterial species with known genomes (*E. vietnamensis*, *S. oneidensis*, *E. coli*, *M. ruber*, and *P. putida*) (*Table 1*; Materials and methods). We first used a dilution of the control sample at ~10 cells per sub-sample. Then, we further diluted the control sample so that each sub-sample contained 0.5 cells on average, effectively performing microfluidic single-cell MDA with the same downstream steps (Materials and methods). Performing MDA from multiple cells in a 300 nL micro-fluidic chamber improved genome coverage at similar sequencing depths because smaller MDA reaction volumes tend to reduce amplification gain and bias (*de Bourcy et al., 2014*; *Zong et al., 2012*).

Thirty mini-metagenomic and 36 single-cell limiting dilution sub-samples yielded 169 and 194 million paired end reads, respectively. Initial trimming removed 14 ± 1.1% (s.d.) reads; 80 ± 1.5% (s.d.) of the remaining reads mapped uniquely to reference genomes (*Figure 1—figure supplement 2A*). Unmapped reads made up only 3% of all reads (*Figure 1—figure supplement 2B*). Improperly mapped reads (3 ± 0.9% s.d.) were mostly short, low quality sequences, and chimeras represented <1% of the reads. Mini-metagenomic MDA reactions produced higher median coverage than single-cell MDA reactions at all sequencing depths (*Figure 1—figure supplement 3A*). The proportion of assembled genome as a function of aligned genome, however, did not differ significantly between mini-metagenomic and single-cell methods (*Figure 1—figure supplement 3B*). Therefore, data from mock bacterial communities demonstrate that with the mini-metagenomic method, less sequencing cost is required to recover similar genome coverage as compared to single-cell experiments, with the improvement mostly associated with the amplification rather than assembly steps.

## Microfluidic-based mini-metagenomics enables contig binning based on co-occurrence patterns

Next, we performed microfluidic mini-metagenomic sequencing on two hot spring samples from Yellowstone National Park (*Figure 2*; Materials and methods). Sample #1 was collected from Bijah Spring and sample #2 was collected from Mound Spring (*Table 2*). 121 and 133 million paired end reads were obtained from the two samples respectively. We removed sub-samples with less than 800,000 paired end reads, yielding 49 and 93 sub-samples respectively. After quality filtering, >90% reads from each sub-sample were incorporated into contigs during independent assemblies (*Figure 2—figure supplement 1*). Re-assembling combined sub-sample reads increased contig length (*Figure 2—figure supplement 2A*). We obtained 643 and 1474 contigs longer than 10 kbp from the two samples, respectively, and used these contigs for subsequent analyses (*Figure 2—figure supplement 2B*; *Table 3*). To compare this performance to shotgun metagenomic assemblies, we obtained 32.5 million and 51.4 million shotgun metagenomic reads from Bijah Spring and Mound Spring samples and down sampled combined mini-metagenomic reads to match shotgun sequencing depths. After assembly (Materials and methods), we observed that for the Bijah Spring sample, where 49 sub-samples were available, the shotgun metagenomic assembly produced more contigs over 10 kbp (*Figure 2—figure supplement 3A*). However, for the Mound Spring sample, where 93 sub-samples were used, the mini-metagenomic assembly produced more contigs over 10 kbp. The largest contig from the mini-metagenomic assembly was also longer for the Mound Spring sample (*Figure 2—figure supplement 3B*). Therefore, it is likely that increasing the number of mini-metagenomic sub-samples improves assembly performance compared to shotgun metagenomics. It is also possible that the observed assembly improvement is due to higher complexity of the Mound Spring

**Table 1.** Species used to construct mock bacterial communities.

| Species name | GC content (%) | Culture source | Growth medium | Assession numbers |
|---|---|---|---|---|
| *Echinicola vietnamensis* KMM6221 | 44.7 | DSM 17526 | Marine broth | NC_019904.1 |
| *Shewanella oneidensis* MR1 | 45.9 | JGI | LB | NC_004347.2 |
| *Escherichia coli* MG1655 | 50.8 | ATCC 700926 | LB | NC_000913.3 |
| *Pseudomonas putida* F1 | 61.9 | ATCC 700007 | Nutrient medium | NC_009512.1 |
| *Meiothermus ruber* | 63.4 | DSM 1279 | Termus ruber medium | NC_013946.1 |

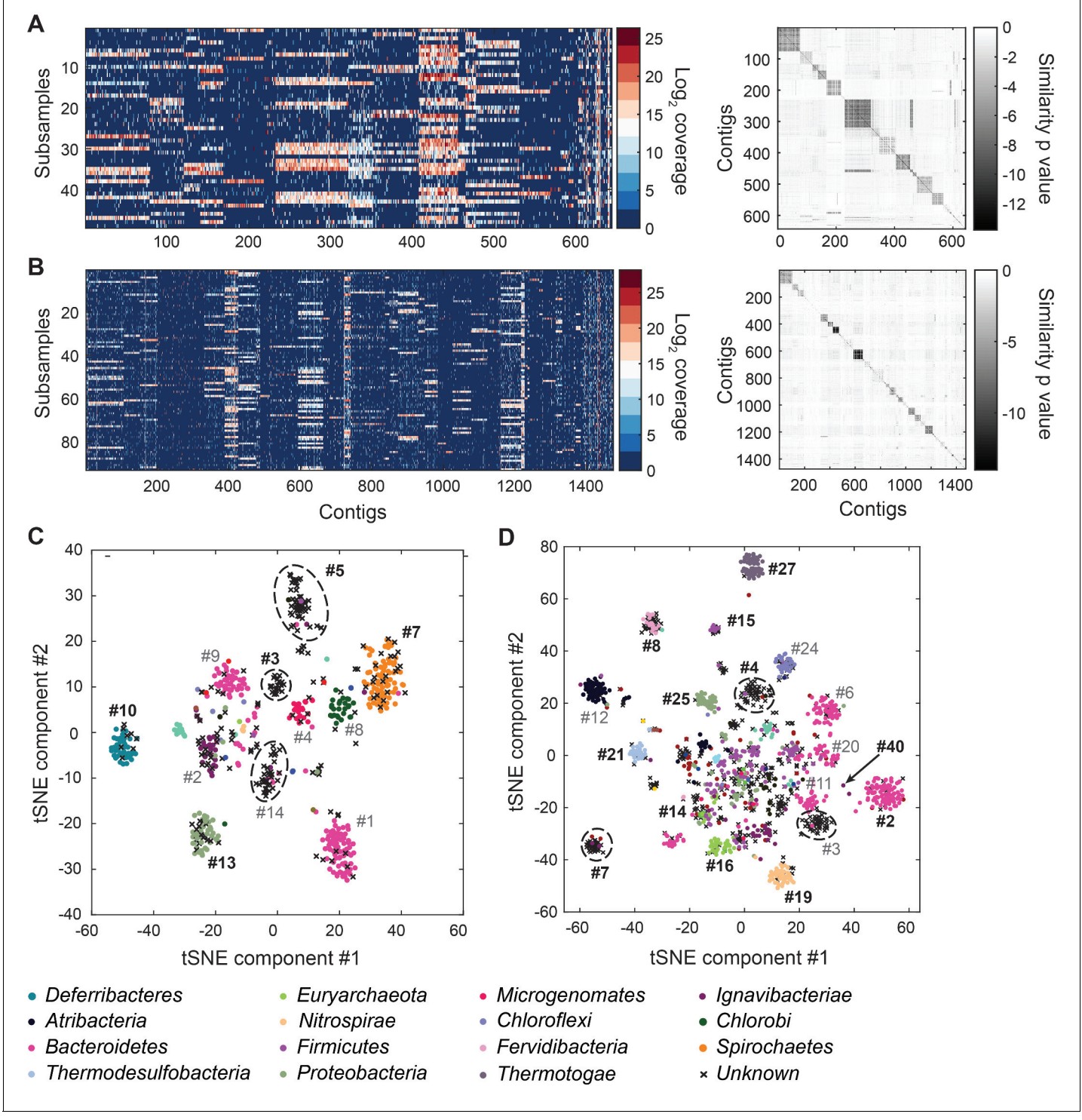

**Figure 2.** Genome bins extracted from microfluidic-based mini-metagenomic sequencing of Yellowstone National Park samples. Two samples from Bijah Spring in Mammoth Norris Corridor (**A, C**) and Mound Spring in Lower Geyser Spring Basin (**B, D**) were collected from Yellowstone National Park and analyzed using the microfluidic-based mini-metagenomic pipeline. (**A, B**) Heat maps of contig coverage across sub-samples are clustered hierarchically to reveal contigs that appear in similar sets of sub-samples (left). Colors represent logarithm of coverage in terms of number of base pairs in base 2. Pairwise *p* values generated using Fisher's exact test based on co-occurrence pattern of contig pairs reveal contig clusters (right). Shading here represents logarithm of *p* value in base 10 after correcting for multiple comparisons. (**C, D**) tSNE dimensionality reduction generated from pairwise *p* values. Each point represents a 10 kbp or longer contig. Colors represent assignment of each contig to a particular phylum based on annotation of genes on the contig. Black X's represent contigs unable to be assigned to any phylum because too many genes have unknown annotation. Genome

*Figure 2 continued on next page*

*Figure 2 continued*

bins larger than 0.5 Mbp are numbered and those with substantial numbers of single-copy marker genes for incorporation into *Figure 4* are labeled in bold. Dotted circles outline genomes predominantly containing contigs unassigned at the phylum level.

The following figure supplements are available for figure 2:

**Figure supplement 1.** Mapping rate of mini-metagenomic sequences.

**Figure supplement 2.** Contig statistics of Yellowstone National Park samples.

**Figure supplement 3.** Comparison between contig statistics of mini-metagenomic and shotgun metagenomic assemblies.

**Figure supplement 4.** An example of computing *p* values using Fisher's exact test and presence patterns of two contigs.

**Figure supplement 5.** DBscan clustering of mini-metagenomic contigs.

**Figure supplement 6.** Gene count as a function of contig length.

bacterial community, validating our hypothesis that the mini-metagenomic method benefits the analysis of complex microbial populations.

Another advantage of mini-metagenomics lies in the information from sub-samples. In order to bin contigs into genomes, reads from each sub-sample were aligned back to mini-metagenomic contigs and coverage was tabulated (*Figure 2A,B*, *Figure 2—figure supplement 2C,D*). Many contig sets share similar coverage patterns across sub-samples, suggesting that they originate from the same genome. Because MDA results in variable coverage profiles, we generated a binary occurrence map by applying a coverage threshold (Materials and methods). Pairwise *p* values were computed with Fisher's exact test (*Figure 2A,B*, *Figure 2—figure supplement 4*). Finally, dimensionality reduction using pairwise *p* values as a distance metric generated clusters of contigs belonging to the same genomes (*Maaten and Hinton, 2008*) (*Figure 2—figure supplement 5*).

To verify the validity of presence-based contig clusters, we annotated all predicted open reading frames (ORFs) (*Huntemann et al., 2016*) (Materials and methods). The relationship between contig length and the number of genes found is linear (*Figure 2—figure supplement 6*), consistent with small non-coding regions in bacterial genomes. Annotations were found for ~50% of the genes, and lineage assignment was performed using JGI's IMG/ER pipeline for metagenome annotation based on gene annotations from the same contig (*Huntemann et al., 2016*). If protein sequences of most genes on a contig were distantly related to known sequences, the contig was designated as unknown (*Figure 2C,D*) (Materials and methods). Approximately 70% of all contigs were assigned at the phylum level. Contigs with known assignment and from the same cluster always belonged to the

**Table 2.** Information of hot spring samples from Yellowstone National Park.

|  | Sample #1 | Sample #2 |
| --- | --- | --- |
| NPS study number | YELL_05788 | YELL_05788 |
| Sample area | Mammoth Norris Corridor | Lower Geyser Basin |
| Location name | Bijah Spring | Mound Spring |
| Sample type | Sediment | Sediment |
| Collection time | September 11, 2009 5:00pm | September 14, 2009 2:25pm |
| Location | 44.761133 N, 110.730900 W | 44.564833 N, 110.859933 W |
| Location type | Hot spring | Hot spring |
| Temperature (°C) | 65 | 55 |
| pH | 7.0 | 9.0 |

**Table 3.** Yellowstone sample sequencing and assembly statistics.

|  | Sample #1 | Sample #2 |
| --- | --- | --- |
| IMG genome ID | 3300006068 | 3300006065 |
| Number of molecules sequenced | 120.5 M | 133.1 M |
| Number of contigs (>10 kbp) | 643 | 1474 |
| Number of subsamples analyzed | 49 | 93 |

same phylum, demonstrating that the presence-based method can correctly bin metagenomic contigs into genomes (*Figure 2C,D*). Some unassigned contigs are scattered among genome bins with known phylum level assignments and likely represent novel genes. Genome bins containing predominantly unassigned contigs are indicated by dotted circles (*Figure 2C,D*) and likely represent deeply branching lineages.

We selected 29 partial genome bins from both hot spring samples with genome sizes over 0.5 Mbp for downstream analyses (*Figure 3A*). Assessment of genome bins demonstrated various levels of completeness and <5% marker gene duplication (*Figure 3B,C*). Genome completeness was not necessarily correlated with genome size because completeness was assessed through single-copy marker genes using CheckM (*Parks et al., 2015*). Using lineage assignment based on gene annotations, we identified eight genome bins with known phylum level assignments from the Bijah Spring sample. Three genomes (Bijah #3, Bijah #5, and Bijah #14) had unknown assignments (*Figure 2C*). In the Mound Spring sample, more reads and sub-samples resulted in 14 genomes with known phylum level assignments and three unassigned genomes (Mound #3, Mound #4, and Mound #7) were extracted (*Figure 2D*). A singleton contig 3.3 Mbp in length (#40) was also included as a separate genome. In both environmental samples, the ability to separate distinct clusters of *Bacteroidetes* contigs represents an advantage of our sequence-independent binning approach, where the presence of bacterial species across microfluidic chambers is determined only by Poisson distribution. Hence, the likelihood that closely related bacterial cells were isolated into the same chambers was small.

Since extracted genomes of known phyla often represented novel lineages at lower taxonomic levels, we attempted to identify their phylogenetic placements. Seventeen genomes (shown in bold in *Figure 2C,D*), including four unassigned genomes, were complete enough for phylogenetic tree construction based on 56 single copy marker genes (*Eloe-Fadrosh et al., 2016*) (*Figure 4*; Materials and methods). All marker gene based phylogenetic assignments were consistent with annotation based assignments. We identified 13 genomes with short to medium branch length from known lineages (red), and four deeply branching lineages that may represent potentially novel phyla (red and starred). In addition to bacterial lineages, we also extracted partial archaeal genomes belonging to *Euryarchaeota* (Mound #16) and *Bathyarchaeota* (Mound #14) (*Figure 4*). The remaining two genomes (Bijah #14 and Mound #3) with unassigned phylogeny that were also not complete enough for incorporation into the final phylogenetic tree were examined independently. Phylogenetic trees built from blast results of individual ribosomal protein sequences suggest that both genomes belonged to the phylum *Ignavibacteriae* (*Figure 4—figure supplement 1*, *Figure 4—figure supplement 2*).

## Functional analyses reveal dominant energy metabolism in Yellowstone hot spring samples

To understand more about the metabolism of these organisms, we performed BLAST of all ORFs contained in each genome against the KEGG database and mapped results onto KEGG modules (Materials and methods). *Figure 5* illustrates the proportion of identified KEGG module genes from each genome belonging to pathways associated with nitrogen, methane, and sulfur metabolism. At the community level, the Mound Spring population displayed higher potential for methanogenesis than the Bijah Spring population, and methanogenesis can be carried out by members of the *Euryarchaeota* (Mound # 16) lineages involving *mcr* and *hdr* complex (*Ferry, 2011*). Formaldehyde

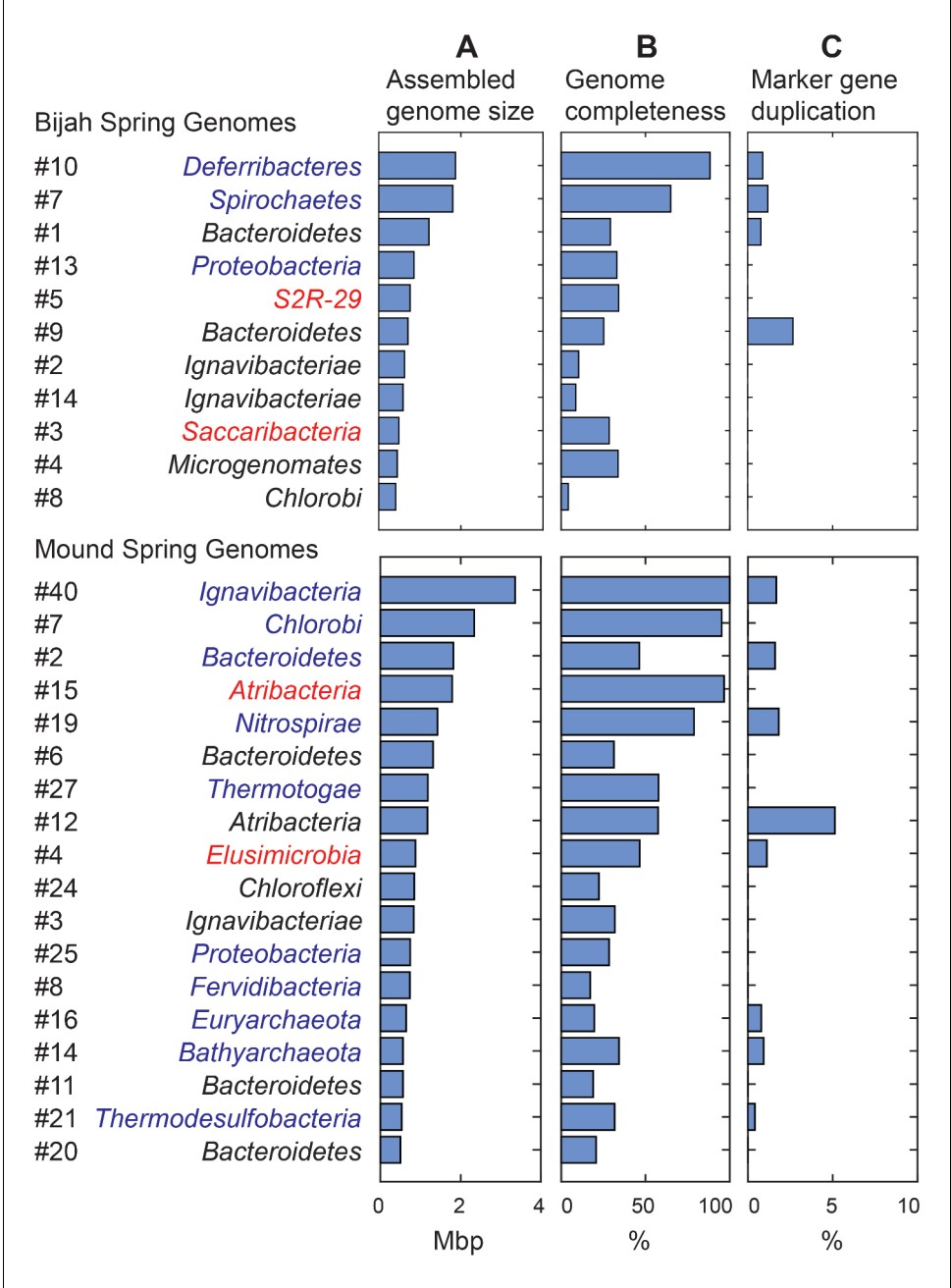

**Figure 3.** Assembled size and completeness of Yellowstone hot spring genomes. Genomes of Bijah Spring and Mound Spring samples are sorted by assembled genome size (**A**). Names represent phylum level assignment based on annotated genes (*Figure 2*), concatenated marker gene phylogenetic tree (*Figure 4*), or individual marker gene trees (*Figure 4—figure supplement 1*, *Figure 4—figure supplement 2*). (**B**) Genome completeness is assessed through single-copy marker genes; those incorporated into *Figure 4* have phyla names colored in blue (for short branching lineages) or red (for deeply branching lineages). (**C**) Degree of marker gene duplication in assembled genomes assessed using CheckM (*Parks et al., 2015*).

assimilation, on the other hand, could be carried out in both communities. In nitrogen metabolism, we identified five genomes including *Ignavibacteriae* (Mound #40), *Bacteroidetes* (Mound #2), *Nitrospirae* (Mound #19), *Thermodesulfobacteria* (Mound #21), and *Ignavibacteriae* (Bijah #2) carrying *nar*, *nrf*, and *nir* genes, possibly participating in the conversion of nitrate to ammonia

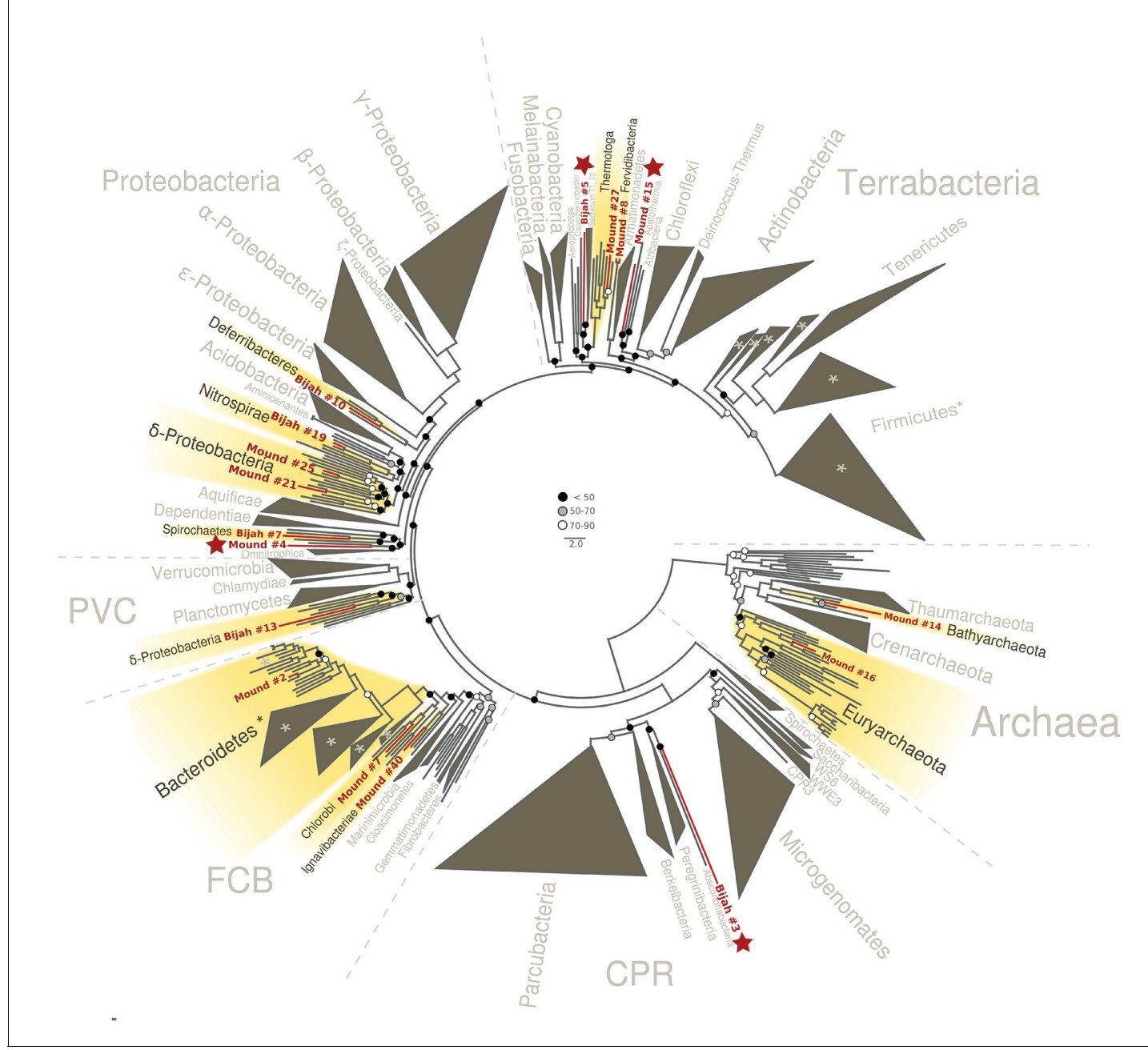

**Figure 4.** Phylogenetic distribution of selected Yellowstone hot spring genomes (red branches) across a representative set of bacterial and archaeal lineages. Query genomes which potentially represent novel phyla are marked with a star, and those falling into known phyla are highlighted in yellow. Bootstrap support values are displayed at the nodes as filled circles in the following categories: no support (black; <50), weak support (grey; 50–70), moderate support (white; 70–90), while absence of circles indicates strong support (>90 bootstrap support). For details on taxon sampling and tree inference, see Materials and methods.

The following figure supplements are available for figure 4:

**Figure supplement 1.** Single gene trees based on multiple sequence alignment of 10 most similar protein sequences for Bijah Spring genome #14 based on NCBI protein blast.

**Figure supplement 2.** Single gene trees based on multiple sequence alignment of 10 most similar protein sequences for Mound Spring genome #3 based on NCBI protein blast.

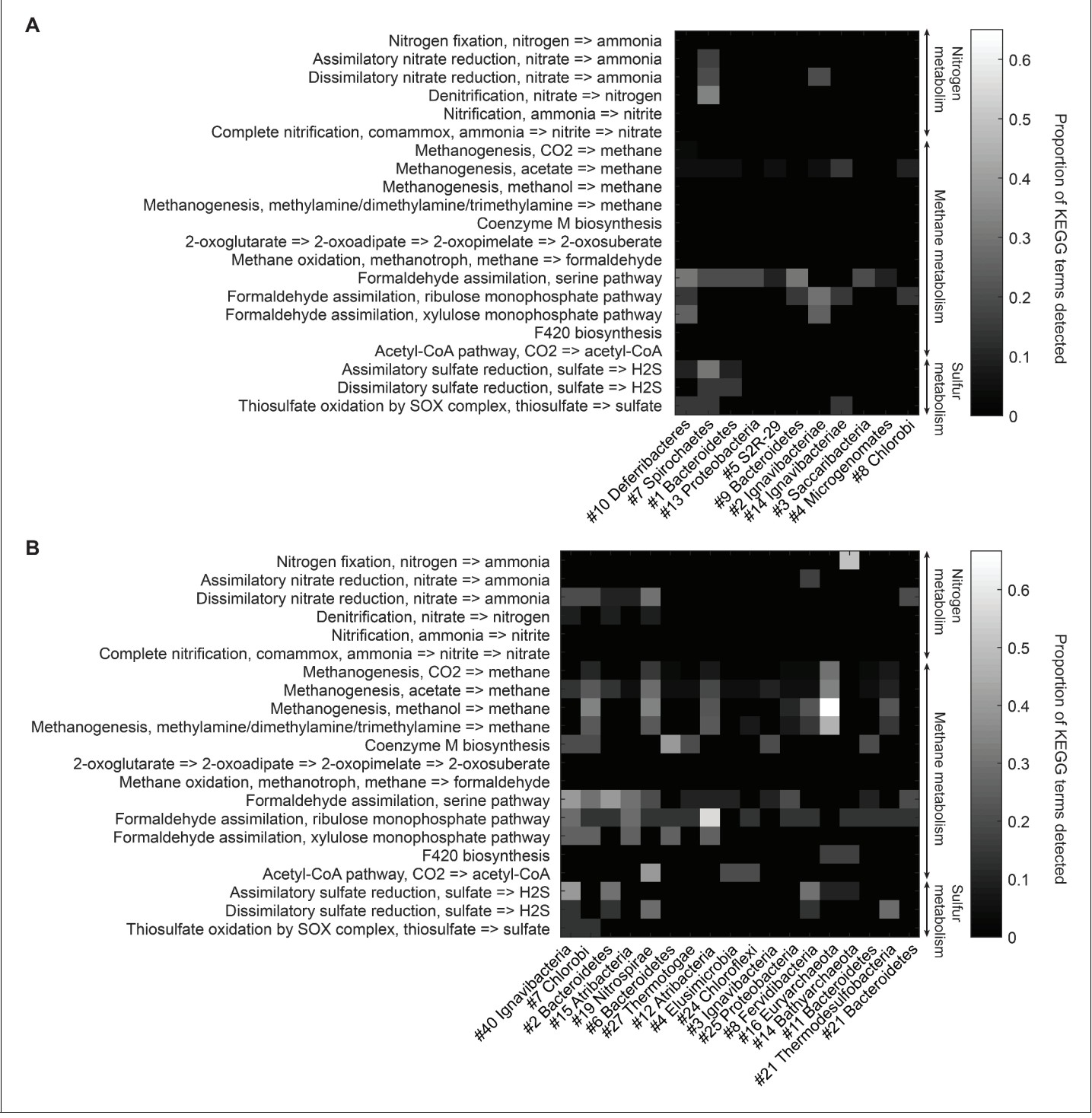

**Figure 5.** Functional analysis of Yellowstone hot spring genomes. Abundant genes involved in energy metabolism of (**A**) Bijah Spring and (**B**) Mound Spring genomes. Each row represents description of a pathway based on KEGG energy metabolism modules. Each column represents one genome bin. Shading of each square represents the ratio of genes in each KEGG module that are also present in a particular genome bin. Modules are labeled as nitrogen metabolism, methane metabolism, or sulfur metabolism.

(*Sodergren and DeMoss, 1988*; *Einsle et al., 1999*; *Cantera and Stein, 2007*). Interestingly, the archaeal genome belonging to *Bathyarchaeota* (Mound #14) carries *nifD* and *nifH*, capable of converting nitrogen directly to ammonia (*Fani et al., 2000*) (*Figure 5B*). Although we did not identify denitrification genes in the Mound Spring community, a genome extracted from Bijah Spring belonging to *Spirochaetes* carried *nirS*, *norBC*, and *nosZ* genes, capable of reducing nitrite to nitrogen (*Figure 5A*).

Bacterial clades associated with thermophilic environments in the Yellowstone National Park are known for their sulfur reduction (converting sulfate to sulfide) activities (*Henry et al., 1994*; *Inskeep et al., 2013*). Two well characterized pathways are assimilatory and dissimilatory processes, where sulfur is either incorporated into cellular materials or serves only as the terminal electron acceptor (*PeckPeck, 1961*). We identified genes associated with the assimilatory sulfate reduction pathway (*cysNC*, *sat*, *cysC*, *cysH*, and *cysI*) from three Mound Spring genomes including *Ignavibacteriae* (Mound #40), *Bacteroidetes* (Mound #2), *Fervidibacteria* (Mound #8) and one Bijah Spring genome – *Spirochaetes* (Bijah #7). In the Mound Spring population, we also identified key genes (*dsrAB*) responsible for dissimilatory sulfate reduction from *Nitrospirae* (Mound #19) and *Termodesulfobacteria* (Mound #21) genomes (*Figure 5*). These insights based on genes with known annotations demonstrate the ability of mini-metagenomics to reveal different metabolic pathways associated with individual genomes within and across environmental samples.

## Microfluidic-based mini-metagenomics facilitates assessment of genome abundance and population diversity with single-cell resolution

Unlike typical shotgun metagenomic methods that use coverage depth as a proxy for genome abundance, mini-metagenomics uses occurrence of genomes across sub-samples to estimate abundance through counting cells. Thresholding coverage depth into an occurrence profile can be seen as digitizing an otherwise noisy analog signal, potentially reducing the effect of amplification bias, copy number variation, and genome size across genomes. Because cells were well-mixed before loading into the Fluidigm C1 IFC and that the microfluidic structures inside each chamber were much larger than the size of a microbial cell, cells were distributed randomly into microfluidic chambers, with occurrence profiles satisfying a Poisson distribution. We quantified the number of detected cells of all genomes in both samples (*Figure 6A,C,D*; Materials and methods). Assembled genomes from the Bijah sample covered one order of magnitude in abundance. Genomes #10 (*Deferribacteres*) and #8 (*Chlorobi*) were most abundant, with 49 and 29 cells represented. On the lower end, Bijah #5 (*S2R-29*) was only represented in four sub-samples, most likely representing only four cells (*Figure 6C*). In the Mound Spring sample, more sub-samples were sequenced, allowing the quantification of a larger abundance range across two orders of magnitude (*Figure 6D*). Mound #40 (*Ignavibacteria*) was the most abundant lineage, appearing in all but seven sub-samples. The abundance of this genome was likely the reason for the successful assembly of its entire genome (*Figure 6B*). Mound #7 (*Chlorobi*) and deeply branching Mound #15 were also abundant, allowing the capture of 36 and 65 cells respectively. For rare organisms, we detected Mound #11 (*Bacteroidetes*) and Mound #14 (*Bathyarchaeota*), which were present at <1%. Comparing with relative abundance computed from shotgun coverage, we observed general agreement in relative abundance profiles, although a couple of genomes in Bijah Springs displayed large differences (*Figure 6—figure supplement 1A*). Relative abundances were higher for less abundant genomes when the mini-metagenomic method was use, demonstrating its sensitivity to extracting rare phylogenies (*Figure 6—figure supplement 1A*). In total, examining genomes larger than 0.5 Mbp, we captured 192 and 509 cells from Bijah and Mound Springs, respectively, corresponding to four to six cells per sub-sample. Even though this number was smaller than our intended ten cells per chamber, the difference was likely due to lysis inefficiency or smaller genomes excluded from our analysis.

In addition to quantifying abundance through counting single cells, genetic variation among individual cells with the same genome can be assessed as well. An examination of species level lineage assignment to contigs revealed that genomes had consistent species level assignments when such assignments were present (*Figure 6—figure supplement 1B*). Therefore, even though genomes were generated from a collection of cells across sub-samples, these cells represented genetic lineages at the species levels. We quantified observed single nucleotide polymorphisms (SNPs) in each assembled genome normalized to the total assembled genome size (*Figure 6B*; Materials and methods). We found a wide distribution in SNP abundance among phylogenies. All *Ignavibacteriae* and

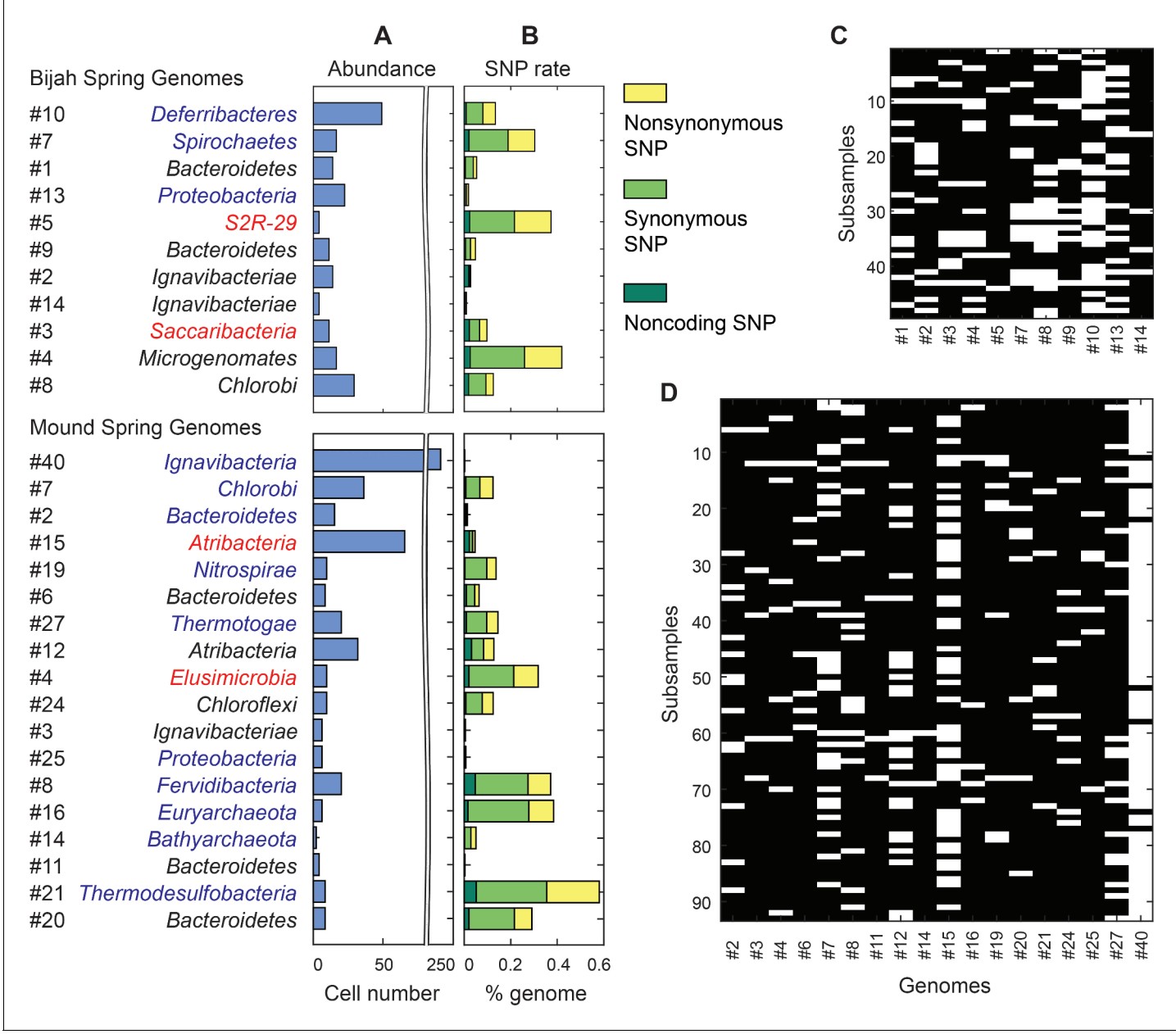

**Figure 6.** Abundance and population variation of Yellowstone hot spring genomes. (A) Abundance is derived from the occurrence pattern of contig clusters, where Poisson distribution is used to infer the number of cells processed. (B) SNPs are tabulated and normalized by the total size of sequenced genome. Most SNPs are in coding regions of the genome, of which the majority are synonymous. (C, D) Map of genome occurrence patterns across all sub-samples for (C) Bijah and (D) Mound Springs samples. White demonstrates the presence of at least one cell of a particular genome in a sub-sample. The total number of cells can be inferred using Poisson statistics.

The following figure supplements are available for figure 6:

**Figure supplement 1.** Genome relative abundance and taxonomic specificity.

**Figure supplement 2.** Observed SNP rates compared to other variables.

three *Bacteroidetes* genomes have low SNP rates (<1%). In the Bijah Spring sample, members of *Spirochaetes*, *Microgenomates*, and *S2R-29* contain the most SNPs at 0.3–0.4% of the assembled genome. In the Mound Spring sample, members of *Thermodesulfobacteria* have the most SNPs (0.6%). Several other groups belonging to *Bacteroidetes*, *Fervidibacteria*, *Euryarchaeota*, and deeply branching Mound #4 all have a 0.3–0.4% SNP rate. It is not the case that larger, more complete, or more abundant genomes contain more SNPs in a population (*Figure 6—figure supplement 2*), indicating that the observed SNP rate is a biological property of the particular genome. The ratio of nonsynonymous to synonymous SNP rates (dN/dS) is typically a measure of the strength of environmental selection (*Holt et al., 2008*), with one indicating no selection and zero indicating strong selection. Among diverse genomes that allow more accurate assessment of SNP ratios, we found dN/dS ranging from 0.4 to 0.9 in the hot spring genomes, illustrating that different phylogenies in the same hot spring environment are subjected to different weak to moderate negative selection pressures. Taken together, the microfluidic-base mini-metagenomic method creates the statistical power from multiple sub-samples to more effectively bin contigs for functional analysis, quantify abundance, and assess genomic variation.

## Discussion

Previous work has implemented similar concepts of sequencing multiple bacterial cells together in order to increase throughput. However, the authors treated each sub-sample as an independent entity throughout their bioinformatic analysis (*McLean et al., 2013*). The novelty of our analysis derives from treating information from sub-samples as different but overlapping sections of a more complex metagenome. Such an approach echoes recent work that uses coverage from many shotgun metagenomic samples containing a similar set of microbial phylogenies but with different abundances in order to aid bioinformatic genome binning (*Alneberg et al., 2014*; *Nielsen et al., 2014*). Our approach differs with these methods in several aspects. First, our approach generates sub-samples from one, possibly low volume, microbial sample, eliminating the need to collect multiple spatial or temporal metagenomic samples. In addition, because of low complexity, each sub-sample does not need to be sequenced as deeply as using shotgun metagenomic methods, significantly saving sequencing cost. Although each sub-sample queries only a small subset of the true diversity, summation of all sub-samples approximates the original environment. Because of MDA bias, we do not use coverage depth directly for contig binning. Instead, as noted previously, thresholding coverage depth into an occurrence profile digitizes a possibly noisy analog signal. Such digitization reduces sensitivity to amplification and other bias associated with analog coverage signals.

Microfluidic-based mini-metagenomics provides researchers with two independent experimental design parameters in order to optimize the protocol for microbial samples of different complexities: the average number of cells per microfluidic chamber and the number of chambers to sequence. The statistical power of the presence based binning technique benefits from both the presence and absence of genomes across sub-samples. Therefore, for less complex samples, reducing the number of cells per microfluidic chamber ensures that cells of the same lineage do not appear in all sub-samples. For more complex samples, the number of cells per chamber can be increased to capture higher diversity. One can even carry out multiple runs with different loading densities to tackle both abundant and rare species. Mini-metagenomics uses a commercially available instrument, thereby facilitating its adoption by groups without expertise in microfluidic technologies. More importantly, because microfluidic-based mini-metagenomics provides orthogonal information to tetranucleotide or coverage information derived from traditional metagenomic sequencing, combining these approaches represents an exciting future direction for deriving a more comprehensive picture of the microbial world.

## Materials and methods

### Mock sample construction

The artificial bacterial community used to test the mini-metagenomic approach was constructed using five model species with different GC content provided by the Joint Genome Institute (*Table 1*). Each species was cultured in test tubes independently in their respective media until saturation

(*Table 1*). Then, each culture was re-suspended in 1% NaCl. A rough cell count was performed on a hemocytometer under an inverted bright field microscope (Leica DMI 6000). Approximately equal number of cells were combined to create an artificial mixed population. Ultrapure glycerol (Invitrogen) was immediately added to the mixture at 30% and stored at −80°C. At the same time, cultures were serially diluted and plated on 2% agar pad containing respective media for each species. After culturing, colonies were counted to get a more accurate quantification of species abundance in the mixed mock community.

## Environmental sample collection and storage

The environmental samples used in this study were collected from two separate hot springs in Yellowstone National Park under permit number YELL-2009-SCI-5788 (*Table 2*). Sample #1 was collected from sediments of the Bijah Spring in the Mammoth Norris Corridor area. Sample #2 was collected from sediment near Mound Spring in the Lower Geyser Basin region. Samples were placed in 2 mL tubes without any filtering and soaked in 50% ethanol onsite. After mixing with ethanol, samples were kept frozen until returning from Yellowstone to Stanford, at which time tubes containing the samples were transferred to −80°C for long term storage.

## Sample preparation and dilution for mini-metagenomic pipeline

Each mock sample was thawed on ice and centrifuged at 5000 x g for 10 min at room temperature. Supernatant was removed and cells were re-suspended in 1% NaCl. Each sample from Yellowstone was also thawed on ice. The tube was vortexed briefly to suspend cells but not large particles and debris. 1 mL of sample from the top of the tube was removed, placed in a new 1.5 mL tube, and spun down at 5000 x g for 10 min to pellet the cells. Supernatant was removed and cells were re-suspended in 1% NaCl. After resuspension, cell concentration was quantified using a hemocytometer under bright field and phase microscopy (Leica DMI 6000). Each sample was then diluted in 1% NaCl or PBS to a final concentration of ~$2\times10^6$ cell/mL, corresponding to ~10 cells per chamber on the Fluidigm C1 microfluidic IFC (Integrated Fluidic Circuit).

## Microfluidic genomic DNA amplification on Fluidigm C1 auto prep system

Because of the small amount of DNA associated with 5–10 cells, DNA contamination was a concern for MDA reactions. To reduce DNA contamination, we treated the C1 microfluidic chip, all tubes, and buffers under UV (Strategene) irradiation for 30 min following suggestions of Woyke et al (*Woyke et al., 2011*). Reagents containing enzymes, DNA oligonucleotides, or dNTPs were not treated. After UV treatment, the C1 IFC was primed following standard protocol (https://www.fluidigm.com/binaries/content/documents/fluidigm/resources/c1-dna-seq-pr-100-7135/c1-dna-seq-pr-100-7135/fluidigm%3Afile). Priming the C1 IFC involved filling all microfluidic control channels with C1 Harvest Reagent, all capture sites with C1 Blocking Reagent, and the input multiplexer with C1 Preloading Reagent. These reagents were available in all C1 Single-Cell Reagent Kits. The diluted environmental sample was loaded onto the chip using a modified version of the loading protocol where washing was not performed, as the capture sites were too large for microbial cells. Hence, they acted essentially as chambers into which cells were randomly dispersed (*Figure 1—figure supplement 1B*). Following cell loading, whole genome amplification via MDA was performed on-chip in 96 independent reactions. A lysozyme (Epicenter) digestion step was added before alkaline denaturation of DNA. After alkaline denaturation of DNA, neutralization and MDA were performed (Qiagen REPLI-g single cell kit) (*Figure 1—figure supplement 1C*). Concentrations of all reagents were adjusted to match the 384 well plate-based protocol developed by the single-cell group at DOE's Joint Genome Institute but adapted for volumes of the Fluidigm C1 IFC (*Rodrigue et al., 2009*). Lysozyme digest was performed at 37°C for 30 min, alkaline denaturation for 10 min at 65°C, and MDA for 2 hr and 45 min at 30°C. The detailed custom C1 scripts for cell loading, DNA amplification, and associated protocols including reagent compositions are available through Fluidigm's ScriptHub at the following link https://www.fluidigm.com/c1openapp/scripthub/script/2017-02/mini-metagenomics-qiagen-repli-1487066131753-8.

## DNA quantification, library preparation and sequencing

Amplified genomic DNA from all sub-samples was harvested into a 96 well plate. The concentration of each sub-sample was quantified in independent wells using the 96 capillary version of the Fragment Analyzer from Advanced Analytical Technologies Inc. (AATI). Because of the size of recovered DNA, we used the high sensitivity large fragment analysis kit (AATI) and followed its standard protocol. The instrument effectively runs 96 independent gel electrophoresis assays, producing an electropherogram for each sub-sample. Using a DNA ladder with known concentrations as reference, the instrument's software quantifies DNA concentration for each sub-sample by performing smear analysis between user specified ranges. For MDA amplified microbial genomic DNA harvested from Fluidigm's C1 IFC, one large smear was often present between 1.5 kbp and 30 kbp. Following quantification, DNA from each sub-sample was diluted to 0.1–0.3 ng/µL, the input range of the Nextera XT library prep pipeline. Nextera XT V2 libraries (Illumina) were made with dual sequencing indices, pooled, and purified with 0.75 volumes of AMpure beads (Agencourt). Illumina Nextseq (Illumina) $2 \times 150$ bp sequencing runs were performed on each library pool.

## Contig construction

A custom bioinformatic pipeline was used to generate combined biosample contigs (*Yu, 2017*). Sequencing reads were filtered with Trimmomatic V0.30 in paired end mode with options 'ILLUMINACLIP:adapters.fa:3:30:10:3:TRUE SLIDINGWINDOW:10:25 MAXINFO:120:0.3 LEADING:30 TRAILING:30 MINLEN:30' to remove possible occurrences of Nextera indices and low quality bases (*Bolger et al., 2014*). Filtered reads from each sub-sample were clustered using DNACLUST, with k = 5 and a similarity threshold of 0.98, in order to remove reads from highly covered regions (*Ghodsi et al., 2011*). Then, assembly was performed using SPAdes V3.5.0 with the sc and careful flags asserted (*Bankevich et al., 2012*). From all sub-sample assembly output, corrected reads were extracted and combined. The combined corrected reads from all sub-samples were assembled again via SPAdes V3.5.0 with kmer values of 33,55,77,99. Finally contigs longer than 10 kbp were retained for downstream analyses.

## Gene annotation

Contigs were uploaded to JGI's Integrated Microbial Genomes's Expert Review online database (IMG/ER). Annotated was performed via IMG/ER (*Huntemann et al., 2016*). Briefly, structural annotations were performed to identify CRISPRs (pilercr), tRNA (tRNAscan), and rRNA (hmmsearch). Protein coding genes were identified with a set of four *ab initio* gene prediction tools: GeneMark, Prodigal, MetaGeneAnnotator, and FragGeneScan. Finally, functional annotation was achieved by associating protein coding genes with COGs, Pfams, KO terms, EC numbers. Phylogenetic lineage was assigned to each contig based on gene assignment.

## Contig co-occurrence distance score and binning

Corrected reads from each sub-sample were aligned back to assembled contigs over 10 kbp using Bowtie2 V2.2.6 with options '–very-sensitive-local -I 0 -X 1000' (*Langmead and Salzberg, 2012*). Total coverage in terms of number of base pairs covered for every contig from every sub-sample was tabulated and subjected to a log transform in base two (*Figure 2A,B*). A threshold value of $2^{11}$ was used to determine if a contig had significant presence in a sub-sample. The sensitivity of the contig binning results to this threshold was low, with threshold values of $2^9$–$2^{13}$ producing similar results. After thresholding contig coverage, the occurrence pattern of all contigs across sub-samples was obtained. Based on co-occurrence patterns, a confidence score for each pair of contigs was computed based on Fisher's exact test (*Figure 2A,B*). This score represented the probability of incorrectly rejecting the null hypothesis that two contigs displayed a particular co-occurrence pattern by chance and can be interpreted as the likelihood of two contigs belonging to cells of the same genome, with lower values increasing this likelihood. Below, we demonstrate an example of how to calculate a similarity *p* value based on co-occurrence pattern of two contigs across a set of sub-samples in *Figure 2—figure supplement 4*. For every pair of contigs X and Y, the null hypothesis states that their co-occurrence patterns are not correlated with each other. Then, we tabulate four values A, B, C, D as shown below.

A = number of sub-samples where X is present and Y is present.

B = number of sub-samples where X is absent and Y is present.
C = number of sub-samples where X is present and Y is absent.
D = number of sub-samples where X is absent and Y is absent.
To test the null hypothesis, we compute a *p* value according to the following equation

$$p = \frac{\binom{a+b}{a}\binom{c+d}{c}}{\binom{a+b+c+d}{a+c}} = \frac{(a+b)!\,(c+d)!\,(a+c)!\,(b+d)!}{a!\,b!\,c!\,d!\,(a+b+c+d)!}$$

From the simplified example, we see that X1 and Y1 are both present in eight sub-samples, hence A = 8. In addition, B = 1, C = 2, D = 6, yielding p=0.013 (*Figure 2—figure supplement 4A*). Since *p* is small, we reject the null hypothesis and conclude that X1 and Y1 are correlated. On the other hand, co-occurrence patterns of X2 and Y2 produces p=0.363, which is not small enough to reject the null hypothesis (*Figure 2—figure supplement 4B*).

Tabulating all pairwise *p* values, we created a similarity matrix containing values between 0 and 1, which also acted as a pairwise distance metric. Using this pairwise distance metric, we performed dimensionality reduction of all contigs using tSNE (*Maaten and Hinton, 2008*) (*Figure 2C,D*, *Figure 2—figure supplement 4*).

## Phylogenetic lineage construction

A set of 56 universal single copy marker proteins (*Supplementary file 1*) was used to place novel genomes into a phylogenetic tree together with a representative set of bacterial and archaeal reference genomes. Marker proteins were identified with hmmsearch (version 3.1b2, hmmer.org) using a specific hmm for each of the markers. For every protein, alignments were built with MAFFT (*Katoh and Standley, 2013*) (v7.294b) using the local pair option (mafft-linsi) and subsequently trimmed with trimAl 1.4 (*Capella-Gutiérrez et al., 2009*), removing sites for which more than 90 percent of taxa contained a gap. Query genomes lacking a substantial proportion of marker proteins (less than 28) or which had additional copies of more than one single-copy marker were removed from the data set. In total 17 of 29 Yellowstone hot spring genomes contained sufficient number of marker genes to be included in the tree. Single protein alignments were then concatenated resulting in an alignment of 51,239 sites. Maximum likelihood phylogenies were inferred with ExaML (*Kozlov et al., 2015*) (version 3.0) using the GAMMA model for rate heterogeneity among sites and the LG substitution matrix (*Le and Gascuel, 2008*) and 300 non-parametric bootstraps. The resulting phylogenetic tree was visualized in ete3 (*Huerta-Cepas et al., 2016*).

## Comparison to shotgun metagenomics

Bulk genomic DNA were extracted from Yellowstone National Park hot spring samples using Qiagen's blood and tissue kit using the protocol for DNA extraction from gram positive bacteria. Nextera V2 libraries were constructed and sequenced on Illumina's Nextseq platform. 32.5 million and 51.4 million reads were obtained from Bijah Spring and Mound Spring samples and trimmed using the same parameters as the mini-metagenomic sequencing reads. Finally, assembly is performed using Megahit (*Li et al., 2015*), with default options and kmer values of 21, 31, 41, 51, 61, 71. At the same time, we combined all mini-metagenomic reads from sub-samples and down sampled to the same depth randomly as the shotgun metagenomic sequencing experiments and performed re-assembly using SPAdes.

## Analysis of genes involved in energy metabolism

From each genomic bin, ORFs assigned KO terms during the annotation process were mapped to all KEGG module involved in energy metabolism. For each KEGG module, we counted the number of KO terms extracted from a particular genome as a ratio of all KO terms present in the module. We did not normalize for genome size or completeness because doing so would artificially increase the importance of genes identified from smaller genomes.

## Assessment of genome completeness and abundance

Genome completeness and marker gene duplication (incorporation of contigs that may not belong to the genome) were quantified via CheckM (*Parks et al., 2015*). Genome abundance was quantified using contig presence patterns. If more than 50% of the contigs from a particular genome bin were supported by at least one read in a sub-sample, we concluded that there was at least 1 cell in that sub-sample. Otherwise, zero cells were present in that sub-sample. Because a well-mixed cell suspension was originally loaded into the C1 IFC, cells were distributed into these microfluidic chambers (which we call sub-samples) randomly and independently. Hence, the problem of quantifying genome abundance can be simplified into the problem of inferring the total number of randomly distributed cells in all **n** sub-samples given that **k** sub-samples contained zero cells. This problem calls for the use of the Poisson distribution. Based on Poisson distribution, assuming we know **X**, the total number of cells with a particular genome contained in all **n** sub-samples, the expected number of sub-samples **k** without cells of that genome is denoted by

$$\frac{k}{n} = e^{-\frac{X}{n}}$$

Then for each genome, the total number of cells **X** contained in all **n** sub-samples can be computed with the following expression.

$$X = -n \times \ln\frac{k}{n}$$

In our experiment, for each genome, the number of sub-samples containing zero cells representing that particular genome was tabulated and the most probable number of cells sampled in the microfluidic device was computed via the equation below.

$$Cell\ Number = -\ (Number\ of\ subsamples) \times \ln\left(\frac{Number\ of\ subsamples\ with\ zero\ cells}{Number\ of\ subsamples}\right)$$

## Assessment of genome variation

SNPs (Single Nucleotide Polymorphism) were tabulated for all genomes from cells observed across all sub-samples by first aligning all reads to contigs in genome bins using samtools version 1.3 mpileup functionality with the –g flag (*Li, 2011*). Then, bcftools was used to call SNPs. We used several criteria to ensure confidence of observed SNPs. First, a SNP must have a quality score larger than 180 based on reads from all sub-samples. The results were not affected if we increase the threshold to higher numbers. Second, we required five reads to support each SNP location from a sub-sample in order to determine if the genome recovered from that sub-sample contains the dominant or alternate allele. Finally, we required that the alternate allele appear as the only allele in at least one sub-sample. If a sub-sample was determined to be heterozygous for a particular allele with high confidence, it was counted as one dominant and one alternate allele because it likely resulted from multiple cells from the same microfluidic chamber. Based on ORF predications, we then classified each SNP as noncoding, synonymous, or nonsynonymous.

## Data availability

Fluidigm C1 IFC script and associated protocols for running the IFC and library preparation are available at https://www.fluidigm.com/c1openapp/scripthub/script/2017-02/mini-metagenomics-qiagen-repli-1487066131753-8

Raw sequencing reads are available from NCBI SRA Run Selector under BioProject PRJNA378813.

Assembled and annotated contigs are available at https://img.jgi.doe.gov/mer/ under IMG Genome IDs 3300006068 and 3300006065. Annotations for the genome Mound #40 (*Ignavibacteria*) can also be found under Genome ID 2630969008.

Analysis scripts for contig assembly are available on Github at the following link https://github.com/brianyu2010/Mini-Metagenomic_Analyses (*Yu, 2017*). A copy is archived at https://github.com/elifesciences-publications/Mini-Metagenomic_Analyses.

## Acknowledgements

The authors would like to acknowledge members of the Quake Lab Sequencing Facility including Ben Passarelli, Gary Mantalas, Jennifer Okamoto, and Norma Neff; Department of Energy (DOE) Joint Genome Institute's (JGI) assembly and annotation teams; Rex Malmstrom (JGI); Danielle Goudeau (JGI); Anastasia Nedderton (Stanford); Jon Deaton (Stanford); and NPS staff at YNP for coordinating the sample collection process including Research Coordinator Christie Hendrix. This work is supported by DOE JGI Emerging Technologies Opportunities Program (ETOP) and Templeton Foundation. F.B.Y. is supported by SGF and NSF GRFP. PCB is supported by the Burroughs Welcome Fund via a Career Award at the Scientific Interface. The work conducted by the US Department of Energy Joint Genome Institute, a DOE Office of Science User Facility, is supported under Contract No. DE-AC02-05CH11231.

## Additional information

### Competing interests

SRQ: Professor Stephen Quake is a shareholder of Fluidigm Corporation. The other authors declare that no competing interests exist.

### Funding

| Funder | Grant reference number | Author |
|---|---|---|
| Department of Energy Joint Genome Institute | Emerging Technology Opportunity Program | Feiqiao Brian Yu Stephen R Quake |
| John Templeton Foundation | 51250 | Feiqiao Brian Yu Stephen R Quake |
| Stanford University | Stanford Graduate Fellowship | Feiqiao Brian Yu |
| National Science Foundation | Graduate Research Fellowship Program | Feiqiao Brian Yu |
| Burroughs Wellcome Fund | Career Award at the Scientific Interface | Paul C Blainey |

The funders had no role in study design, data collection and interpretation, or the decision to submit the work for publication.

### Author contributions

FBY, Conceptualization, Software, Formal analysis, Investigation, Visualization, Methodology, Writing—original draft, Writing—review and editing; PCB, Resources, Investigation, Writing—review and editing; FS, Software, Visualization, Writing—original draft, Writing—review and editing; TW, Conceptualization, Supervision, Project administration, Writing—review and editing; MAH, Conceptualization, Supervision, Methodology, Writing—review and editing; SRQ, Conceptualization, Supervision, Funding acquisition, Visualization, Methodology, Writing—original draft, Writing—review and editing

### Author ORCIDs

Feiqiao Brian Yu, http://orcid.org/0000-0003-3416-3046
Frederik Schulz, http://orcid.org/0000-0002-4932-4677
Tanja Woyke, http://orcid.org/0000-0002-9485-5637
Stephen R Quake, http://orcid.org/0000-0002-1613-0809

## Additional files

### Supplementary files

• Supplementary file 1. Table of COG terms used for making marker gene based phylogenetic tree (*Figure 4*).

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
