## [Decision Letter]

Thank you for submitting your article "Microfluidic-based mini-metagenomics enables discovery of novel microbial lineages from complex environmental samples" for consideration by *eLife*. Your article has been favorably evaluated by Wendy Garrett as the Senior Editor and Cameron Thrash (Reviewer #1) as the Reviewing Editor. The following individuals involved in review of your submission have also agreed to reveal their identity: Brett Baker (Reviewer #2) and Steven Giovannoni (Reviewer #3).

In general the reviewers were favorable towards publication of this work, but the reviewers have a number of concerns that they would like to see addressed first.

*Reviewer #1:*

The authors present a novel use of microfluidic cell-sorting of microbial communities to facilitate a "mini-metagenomics" approach. I particularly like the combined use of statistics with physical separation in subsamples to refine the binning procedure. While I think this technique was vetted well and has great potential for future application, there are a number of conceptual and technical details I'd like to see fleshed out prior to publication.

Subsection “Microfluidic-based mini-metagenomics enables contig binning based on co-occurrence patterns”, third paragraph: Please detail the criteria for taxonomic lineage assignment. How many genes per contig were required to have the same taxonomic assignment for that contig to be declared part of that group? At what level of taxonomic assignment did that hold (phylum, class, etc.)? What percent ID was required to "trust" the taxonomic assignment for a given gene?

Subsection “Microfluidic-based mini-metagenomics enables contig binning based on co-occurrence patterns”, last paragraph: Did the single copy marker gene tree agree with the CheckM tree? Did the CheckM tree help with taxonomic assignment for those organisms without sufficient gene-based information and/or those without enough genes to be included in the single copy marker tree?

Subsection “Functional analyses reveal dominant energy metabolism in Yellowstone hot spring samples”, first paragraph: How novel are these discoveries for metabolism of the group, and was any additional investigation done? I know nrfA genes need vetting (Welsh et al. 2014 AEM), so these kinds of claims could use more evidence than just blast-ing to KEGG. Why is it interesting that the *Euryarchaeota* genome has nitrogen fixation genes? Does the genome belong to a subclade where this is unusual?

Subsection “Microfluidic-based mini-metagenomics facilitates assessment of genome abundance and population diversity with single-cell resolution”, first paragraph: How valid is this assumption considering the various shapes and sizes of cells that might be encountered? Please consider and address. Has there been any investigation into whether cell size/morphology biases the microfluidic process?

Subsection “Microfluidic-based mini-metagenomics facilitates assessment of genome abundance and population diversity with single-cell resolution”, first paragraph: Please describe how well the abundance of the organisms in the mini-metagenomics process reflects the abundance of these organisms in the original samples. I believe this should be possible through comparison with the shotgun metagenomics data. This will provide an improved sense of where in the community rank-abundance curve one may hope to explore with the method. It may also reveal if there are biases related to cells from a specific group, which would help answer my preceding questions regarding cell size and morphology in the microfluidics setting.

Subsection “Microfluidic-based mini-metagenomics facilitates assessment of genome abundance and population diversity with single-cell resolution”, last paragraph: How confident are the authors in using SNP determination and dN/dS ratios on genomes that are only binned at the phylum level? In other words, at what level of taxonomic specificity do you believe each genome represents? Species? Strain? Because it seems to me that if the genome represents an amalgamation of data from multiple genetic lineages within a phylum, SNP and dN/dS information could be misleading.

*Reviewer #2:*

The manuscript is well-written. The methods are sound and the results are justified. I think that this approach might be useful in soils, but given that only 29 genomes were recovered from an YNP hot spring, I personally am not convinced there is a cost advantage or an improvement in genomic reconstruction (based on completeness).

However, I don't see any details about how much sequencing was actually done? This needs to be included, or perhaps I missed it? I would suggest doing an average cost/genome and perhaps do a comparison with output from whole-community assembly and binning. I understand the latter will vary considerably depending on the habitat, so perhaps use a complex community where genomes have been reconstructed like Rifle groundwater (Wrighton et al. 2014)? This might make it more convincing that there is a real advantage to this approach.

As an example, we have been able to obtain 57 genomes (>50%) complete from YNP springs from a modest amount of sequencing, 1 lane of HiSeq and 1 lane of MiSeq. Thus, I'm not particularly impressed with the genome completeness of the bins, as most of them are <50% complete.

Overall, as a test of the approach (testing on mixed cultures and natural communities) this paper does well.

Discussion, first paragraph: – Sure you reduce the cost of sequencing, but you are not getting as much genome by subsampling the community. So this statement is misleading.

*Reviewer #3:*

This paper describes a useful re-purposing of a commercially available single-cell genomics technology to produce metagenomic data that is roughly equivalent to a library of single-cell assemblies in quality, despite the limitations of the microfluidics preventing the actual sorting of single-cells. Those interested in the use of single-cell genomics (SCG) for the exploration of microbial communities, may consider this to be a viable and possibly much more practical approach than some SCG methods. In this regard, it is a very useful methods paper that addresses a technical deficit in the ability of microbial ecologists with limited funds, space, or personnel who may wish to perform SCG (or SCG-like) analyses in conjunction with metagenomics.

---

## [Author Response]

*Reviewer #1:*

*The authors present a novel use of microfluidic cell-sorting of microbial communities to facilitate a "mini-metagenomics" approach. I particularly like the combined use of statistics with physical separation in subsamples to refine the binning procedure. While I think this technique was vetted well and has great potential for future application, there are a number of conceptual and technical details I'd like to see fleshed out prior to publication.*

*Subsection “Microfluidic-based mini-metagenomics enables contig binning based on co-occurrence patterns”, third paragraph: Please detail the criteria for taxonomic lineage assignment. How many genes per contig were required to have the same taxonomic assignment for that contig to be declared part of that group? At what level of taxonomic assignment did that hold (phylum, class, etc.)? What percent ID was required to "trust" the taxonomic assignment for a given gene?*

Functional annotation and taxonomic lineage assignment were both performed as part of JGI’s IMG/ER metagenome annotation pipeline. The details of the metagenome annotation pipeline were outlined in Huntemann et al.’s work. In particular, USEARCH 6.0.294 was used to associate genes to KO terms if “there [was] at least 30% identity and at least 70% of the KO gene sequence [was] covered by the alignment”. The top USEARCH hit for each gene was used to assign phylogenetic lineage to contigs. When at least 30% of the genes had USEARCH hits, the assigned phylogenetic lineage was the last common ancestor of USEARCH hits. Therefore, more contigs were unassigned at lower taxonomic levels. Since we used contigs over 10 kbp, most taxonomic assignments were based on contigs with 7 or more genes. We added a statement that taxonomic assignment was performed by JGI’s IMG/ER metagenome annotation pipeline in the third paragraph of the subsection “Microfluidic-based mini-metagenomics enables contig binning based on co-occurrence patterns” with reference to Huntemann et al.’s paper.

*Subsection “Microfluidic-based mini-metagenomics enables contig binning based on co-occurrence patterns”, last paragraph: Did the single copy marker gene tree agree with the CheckM tree? Did the CheckM tree help with taxonomic assignment for those organisms without sufficient gene-based information and/or those without enough genes to be included in the single copy marker tree?*

CheckM was only used in our analysis to assess genome quality and was not used to produce any phylogenetic assignments. The contig and genome phylogenetic assignments were produced by JGI’s IMG/ER metagenome annotation pipeline described by Huntemann et al., except for those genomes with predominantly unassigned contigs. The phylogenetic assignments of 4 unassigned genomes were determined from marker gene based tree (Figure 4) and the other two were determined using selected single marker gene trees (Figure 4—figure supplement 1 and Figure 4—figure supplement 2). The marker gene based phylogenetic tree (Figure 4) was produced independently using 56 universal single copy marker genes and a set of tools described in the “Phylogenetic lineage construction” subsection of Materials and methods.

*Subsection “Functional analyses reveal dominant energy metabolism in Yellowstone hot spring samples”, first paragraph: How novel are these discoveries for metabolism of the group, and was any additional investigation done? I know nrfA genes need vetting (Welsh et al. 2014 AEM), so these kinds of claims could use more evidence than just blast-ing to KEGG. Why is it interesting that the Euryarchaeota genome has nitrogen fixation genes? Does the genome belong to a subclade where this is unusual?*

The putative functional descriptions of prevalent genes are based on blast results from the KEGG database. Although the mini-metagenomic approach was not designed to improve the gene prediction and annotation results, in point of fact the ability to generate longer contigs and contigs from less abundant genomes facilitates the discovery and annotation of genes from rare cells. The comparison between genes found in Bijah Spring and Mound Spring in terms of methane metabolism is novel, especially since such differences could often be attributed to specific genomes. The reviewer also noted our mentioning of *Euryarchaeota* carrying nitrogen fixation genes. We would like to correct this error because in Figure 5, the *Bathyarchaeota* genome (Mound #14) is the one that carries the *nif* genes (subsection “Functional analyses reveal dominant energy metabolism in Yellowstone hot spring samples”, first paragraph). This observation is interesting because *nif* genes are associated with *Euryarchaeota* genomes and *Bathyarchaeota* genomes typically do not carry *nif* genes.

*Subsection “Microfluidic-based mini-metagenomics facilitates assessment of genome abundance and population diversity with single-cell resolution”, first paragraph: How valid is this assumption considering the various shapes and sizes of cells that might be encountered? [Please consider and address] Has there been any investigation into whether cell size/morphology biases the microfluidic process?*

Size limitations or bias can arise when the cells start to approach the smallest channel dimension in size. The particular C1 IFC we used has large capture sites designed for mammalian cells (used to capture cells 17-25 μm in diameter), and the channels are too large to selectively capture microbial (bacterial or archaeal) cells (Figure 7). Because a well-mixed cell suspension was originally loaded into the C1 IFC, the microfluidic “capture sites” acted as chambers where cells were distributed randomly and independently. Hence, the problem of quantifying genome abundance can be simplified into the problem of inferring the total number of randomly distributed cells in all n sub-samples given that k sub-samples contained zero cells. We added text in the first paragraph of the subsection “Microfluidic-based mini-metagenomics facilitates assessment of genome abundance and population diversity with single-cell resolution” to explain justification for using the Poisson distribution to estimate this process. We also included a more detailed description in the “Assessment of genome completeness and abundance” subsection of Materials and methods.

Author response image 1.**DOI:**
http://dx.doi.org/10.7554/eLife.26580.025

*Subsection “Microfluidic-based mini-metagenomics facilitates assessment of genome abundance and population diversity with single-cell resolution”, first paragraph: Please describe how well the abundance of the organisms in the mini-metagenomics process reflects the abundance of these organisms in the original samples. I believe this should be possible through comparison with the shotgun metagenomics data. This will provide an improved sense of where in the community rank-abundance curve one may hope to explore with the method. It may also reveal if there are biases related to cells from a specific group, which would help answer my preceding questions regarding cell size and morphology in the microfluidics setting.*

We added a new Figure 6—figure supplement 1 to compare genome abundance derived from shotgun and mini-metagenomic methods. The original Figure 6—figure supplement 1 was moved to Figure 6—figure supplement 2. Shotgun metagenomic abundance was computed by counting the number of shotgun reads mapped to genomes generated using mini-metagenomics. All abundance profiles were normalized by the total, resulting in a measure of relative abundance. In general, shotgun derived relative abundance patterns matched those derived via mini-metagenomic cell counting. The agreement was better for Mound Spring genomes. For less abundant genomes, the mini-metagenomic method was more sensitive. These were likely due to “lucky” cells that made it into the sub-samples. We also added a short description in the main text describing this comparison (subsection “Microfluidic-based mini-metagenomics facilitates assessment of genome abundance and population diversity with single-cell resolution”, first paragraph).

*Subsection “Microfluidic-based mini-metagenomics facilitates assessment of genome abundance and population diversity with single-cell resolution”, last paragraph: How confident are the authors in using SNP determination and dN/dS ratios on genomes that are only binned at the phylum level? In other words, at what level of taxonomic specificity do you believe each genome represents? Species? Strain? Because it seems to me that if the genome represents an amalgamation of data from multiple genetic lineages within a phylum, SNP and dN/dS information could be misleading.*

We believe that the binned genomes represent single species. To support this claim, we looked at binned genomes from both springs that had more than 50% of the contigs with phylogenetic lineage assignments at the species level (Table below). We see that among these genomes, almost all contigs were assigned to the same species (i.e. contigs were either unassigned at the species level or assigned to the same species). There were two genomes with high levels of specificity at genus level assignments: *Methanobacterium* and *Thermodesulfobacterium*. For these two genomes, no species level assignments existed for most contigs, demonstrating that they likely represented novel species. Most contigs in the other genomes were unassigned at higher taxonomic levels, making it convenient to refer to them by their phylum level assignments. However, based on previous evidence, these genomes should still represent single lineages. Therefore, we believe that the SNP quantification and dN/dS calculations on the genomes are valid. We included information in the following table, in Figure 6—figure supplement 1 and in the last paragraph of the subsection “Microfluidic-based mini-metagenomics facilitates assessment of genome abundance and population diversity with single-cell resolution”.

Genome IDSpecies nameNumber of contigsNumber of contigs with species level assignmentNumber of contigs assigned to the species in column #2Bijah #7*Leptonema illini*976363Bijah #8*Chlorobium sp. GBChlB*282626Bijah #10*Calditerrivibrio nitroreducens*483737Mound #12*Candidatus Caldatribacterium saccharofermentans*666060Mound #19*Thermodesulfovibrio islandicus*543939Mound #27*Thermotoga hypogea*575656Mound #16*Methanobacterium unassigned*402525Mound #21*Thermodesulfobacterium unassigned*333333

We further think that the theoretical basis for the species level specificity of binned genomes extends from the fact that occurrence patterns are generated through DNA sequence alignments, which is more stringent than protein sequence alignments or kmer based binning methodologies.

*Reviewer #2:*

*The manuscript is well-written. The methods are sound and the results are justified.*

*I think that this approach might be useful in soils, but given that only 29 genomes were recovered from an YNP hot spring, I personally am not convinced there is a cost advantage or an improvement in genomic reconstruction (based on completeness).*

*However, I don't see any details about how much sequencing was actually done? This needs to be included, or perhaps I missed it? I would suggest doing an average cost/genome and perhaps do a comparison with output from whole-community assembly and binning. I understand the latter will vary considerably depending on the habitat, so perhaps use a complex community where genomes have been reconstructed like Rifle groundwater (Wrighton et al. 2014)? This might make it more convincing that there is a real advantage to this approach.*

For mini-metagenomic samples from Bijah and Mound Springs, 121 and 133 million paired end reads were obtained, respectively (subsection “Microfluidic-based mini-metagenomics enables contig binning based on co-occurrence patterns”, first paragraph; Table 2). After removing sub-samples with less than 800,000 paired end reads, 49 and 93 sub-samples were left, respectively. For shotgun comparisons, we obtained 32.5 million and 51.4 million reads from Bijah and Mound Springs respectively. Although we did not compare cost/genome between shotgun and the mini-metagenomic approach, we did compare the number of assembled contigs over 10 kbp and the longest contig from mini-metagenomic and shotgun assemblies at the same sequencing depths (see aforementioned paragraph, Figure 2—figure supplement 3), which will enable the reader to estimate cost differences using whatever the current or future costs of sequencing are.

*As an example, we have been able to obtain 57 genomes (>50%) complete from YNP springs from a modest amount of sequencing, 1 lane of HiSeq and 1 lane of MiSeq. Thus, I'm not particularly impressed with the genome completeness of the bins, as most of them are <50% complete.*

One possible reason for the low genome completeness was the requirement that incorporated contigs be longer than 10 kbp – we have been rather conservative in our assembly requirements. This requirement removed a large number of contigs between 5 kbp and 10 kbp that might have contributed to extracted genomes.

*Overall, as a test of the approach (testing on mixed cultures and natural communities) this paper does well.*

*Discussion, first paragraph: Sure you reduce the cost of sequencing, but you are not getting as much genome by subsampling the community. So this statement is misleading.*

We are comparing to coverage based binning from multiple metagenomic samples from the same environment. In such situations, each shotgun metagenomic sample needs to be sequenced deeply in order to bin contigs using information associated with multiple samples. In the mini-metagenomic method, although each sub-sample only contains a subset of lineages, combining sub-samples approximates the original community. We have added a clarification statement in the first paragraph of the Discussion.